# Compliance to Recommendations and Mental Health Consequences among Elderly in Sweden during the Initial Phase of the COVID-19 Pandemic—A Cross Sectional Online Survey

**DOI:** 10.3390/ijerph17155380

**Published:** 2020-07-26

**Authors:** Johanna Gustavsson, Linda Beckman

**Affiliations:** 1Risk and Environmental Studies, Centre for Societal Risk Research Karlstads Universitet, Universitetsgatan 2, 651 88 Karlstad, Sweden; 2Public Health Sciences, Universitetsgatan 2, Karlstads Universitet, 651 88 Karlstad, Sweden; linda.beckman@kau.se

**Keywords:** risk perception, older adults, COVID-19, mental health

## Abstract

Background (1): In the wake of COVID-19, elderly people have been labelled a risk group. As the pandemic is a new crisis in Sweden, we have no knowledge on how this group perceives the information and recommendations being provided. Complying with these recommendations entails physical distancing and, for some, isolation at home. Methods (2): From 16 April to 15 May 2020, we conducted an online survey targeting people aged 70 and older in Sweden (n = 1854). Results (3): A vast majority of the participants find the information and recommendations clear and reliable. Half of the participants report staying at home all the time, and up to half report decreased mental health in terms of, e.g., feeling depressed, having sleeping problems and that isolation makes them feel bad. However, elderly people are not a homogenous group, and there are gender and demographic differences. (4) Conclusion: At this point, we do not know the full extent of the ongoing pandemic, either in terms of duration or in terms of losses. The Swedish model for action on COVID-19 has not included a lock down. However, elderly people seem to comply with recommendations and practice social distancing to a high degree. This might lead to decreased mental health and long-term effects.

## 1. Introduction

In December 2019, a novel coronavirus labeled SARS-CoV2, causing severe respiratory illness, appeared in Wuhan, China. During the first months of 2020, the virus spread and on 11 March 2020, the World Health Organization (WHO) declared the outbreak a pandemic [1]. Early on, diabetes and hypertension comorbidities were identified as risk factors for severe infection [2], and higher age a primary risk factor for death caused by the SARS-CoV2 [3]. Compared to most countries, Sweden has chosen a different path in efforts to minimise the consequences of the SARS-CoV2 outbreak, i.e., the COVID-19 pandemic. When the Swedish authorities declared a general spread of the virus in the middle of March, instead of a general lockdown, authorities appealed to common sense and recommended strict hand hygiene and social distancing for the general population. However, for risk groups, such as the 1.6 million people aged 70 and up in Sweden, the recommendations have been stricter, including a voluntary quarantine and avoiding all social interactions outside the household [4].

The effectiveness of the recommended measures will be determined by how they are received; adequate and accessible information is of great importance, and has shown to have a mitigating effect on mental health consequences during the pandemic [5]. During the pandemic, trust in government authorities has played an important role [6], and the nature of the crisis has also been important as to how it is perceived [7]. 

The recommendations require older individuals to make substantial behavioural changes, and maintaining health behaviours can be challenging, affected by, e.g., level of self-regulation, social pressure and personal resources [8]. In addition, this population has an increased vulnerability to stressful situations [9,10,11,12] that can impact their well-being greatly. Previous research shows that, in general, perceived isolation increases the risk of depression and anxiety [13], making the recommendations on social distancing a major concern [14]. In addition, systematic reviews have concluded that being an elderly woman, compared to being a man [15,16], as well as a lack or loss of close social contacts [16], are risk factors for mental health problems.

In conclusion, the current situation with the COVID-19 pandemic has been shown to affect people’s mental health, both through forced isolation and through concern about the virus [17]. To our knowledge, few studies have focused on the general elderly population, and no studies are yet published in Sweden. Hence, the purpose of this study is to explore the specific situation of people aged 70 and older in relation to COVID-19, more specifically, how people age 70 and older (a) perceive information and act on recommendations about the COVID-19 pandemic and (b) perceive how their mental health is affected by the current situation. In addition, we explore whether there are any gender differences in the above aspects and any differences between single people and those in a relationship.

## 2. Materials and Methods

### 2.1. Participants and Data Collection

An online questionnaire was distributed inviting the target population, i.e., individuals aged 70 and older in Sweden, to answer. Additionally, we chose to include 43 people that answered that they were 69 years old, as they identified themselves as belonging to the risk group. Data between 12 April and 15 May 2020 was used for this study. The research subjects were recruited via Facebook and the Swedish National Pensioners’ Organisation, the largest in Sweden, where the advertisement and link to the survey was shared with a request to participate. In total, 1854 individuals participated (69.4% women).

The questionnaire included 5 questions with 5 to 10 items in each question about perceptions of information concerning COVID-19, how they perceive and comply with the recommendations made towards the risk group to which they belong as well as how they estimate their mental health status during the pandemic situation. To analyse differences in the answers in relation to gender and civil status, these background variables were collected.

### 2.2. Ethical Considerations

The study was approved by the Swedish Ethical Review Authority (No. 2020-01600).

### 2.3. Questionniare and Variables

Initially, we tested the questionnaire on a smaller group of elderly in order to control the conceptual validity, and after that modified some of the questions and wording in order to facilitate comprehension. The background variables included gender (“women”, “men”, “other”, dichotomised into “women” and “men”), civil status (“married”, single”, “cohabitating with somebody” and “in a relationship”, dichotomised into “single” and “not single”).

Variables concerning information, recommendations and mental health: the information concerning the COVID-19 pandemic (a) is sufficient, (b) is clear, (c) from government authorities is reliable and (d) from media is reliable.

In relation to the COVID-19 pandemic, (a) recommendations from government authorities are clear and concise, (b) I take the recommendations from government authorities very seriously, (c) I know how to act to avoid infection, (d) I have changed my daily routines, (e) I avoid public gatherings, (f) I spend all my time at home, (g) I do grocery shopping as usual and (h) I do grocery shopping but at hours when I think no one is there.

In relation to the COVID-19 pandemic, (a) I worry about my health, (b) I worry about my loved ones, (c) my loved ones worry about me, (d) I worry about economic development, (e) the isolation makes me feel bad, (f) I have trouble sleeping, (g) I feel depressed, (h) I have difficulties concentrating and (i) I worry about the consequences to society.

Response alternatives included ”agree completely”, “agree somewhat”, “neither agree/disagree”, “disagree somewhat” and “completely disagree”, and were later dichotomised into “yes” and “no”.

### 2.4. Analyses

All analyses were performed with SPSS v.25. Response rates were calculated according to the number of respondents per response with respect to the number of total responses to a question. Descriptive statistics were calculated to report on participants’ demographic variables. Differences between men and women for questions regarding information, recommendation and mental health were analysed using Chi-square (χ^2^) (categorical variables). All tests were two-tailed, with a significance level of *p* < 0.05.

## 3. Results

### 3.1. Sample

The analysed sample includes 1854 participants, 1272 (69.4%) women and 561 (30.6%) men. The mean age was 74.72 (SD 4.0), and divided into age groups; n = 1058 (57.1%) were 69 to 74 years old, and n = 796 (42.9%) 75 years old and older. Within the analysed sample, 56.9 % were in a relationship (Table 1).

### 3.2. Information about the Pandemic

As shown in Table 2, a vast majority of the participants perceived the general information about the current COVID-19 pandemic (not specifying the source of information) as sufficient (92.2%), and 87% also found the information clear and concise. The information from authorities was considered reliable by 86.6%, compared to the 50.9% who said information from media was reliable. Significantly more women than men reported that they experienced the information to be clear (88.2% and 84.4%, respectively (*p* = 0.031)) and that information from the media was reliable (53.3% and 45.6%, respectively (*p* = 0.003)). In addition, the respondents reported that they preferred to get information from the newspaper (n = 1666, 89.9%), TV (n = 1625, 87.6%), radio (n = 1130, 60.9%), the Internet (n = 1051, 56.7%), via a mobile phone (n = 430, 23.2%) and via a pamphlet in the mailbox (n = 346, 18.4%).

### 3.3. Recommendations about the Pandemic

As shown in Table 3, the majority of respondents reported taking the recommendations of how to act during the COVID-19 pandemic seriously (96.8%). However, women reported taking the information more seriously than men (97.5% and 95.1%, respectively (*p* = 0.006)). Almost all respondents (99.1%) said they avoid public gatherings, and about half of the respondents (48.0%) reported that they did their grocery shopping at hours when they think no one is in the store, with significantly more women than men reporting this (46.1% and 52.2%, respectively (*p* = 0.018)). More than half of the respondents (63.2%) reported that they spend all of their time at home, with a significant larger share of women compared to men reporting so (66.0% and 56.7%, respectively (*p* < 0.000)).

### 3.4. Mental Health during the Pandemic

About half of the respondents (60.8%) said they worry about their health during the COVID-19 pandemic, with significantly more women reporting this (62.3% and 57.2%, respectively (*p* = 0.042)) (Table 4). About 80% worried about their loved ones. When it came to how the respondents felt during isolation, about half of them felt bad, and there were significantly more women reporting negative feelings than men (55.4% and 37.2%, respectively (*p* < 0.000)). Similar gender patterns were further seen in having trouble sleeping, feeling depressed, having difficulties concentrating and worrying about economic downturns and consequences to society, where women reported more negative feelings (*p* < 0.000).

### 3.5. Mental Health and Civil Status

The results in Table 5 shows no difference in self-reported worry about health across those in a relationship and those not. There is significant statistical difference in terms of worry about loved ones, where it is more common to worry about loved ones when in a relationship. It was significantly more common to feel bad from the isolation when single (56.9% compared to 45.3%, *p* < 0.000). Similarly, it was more common to have sleeping problems when single compared to those in a relationship (20.7% and 27.4%, respectively, *p* < 0.000), feeling depressed (33.5% and 46.4%, respectively, *p* < 0.000), and having difficulties concentrating (19.3 and 28.6%, respectively, *p* < 0.000).

## 4. Discussion

### 4.1. Results Discussion

This study explores how older people in the targeted risk group perceive the COVID-19 pandemic in its initial phase in spring 2020 as to how they perceive information, comply with recommendations and how they feel their mental health is being affected in the current situation.

In the efforts to save lives and limit the strain on the healthcare system during the COVID-19 pandemic, compliance to recommendations is vital. Awareness and knowledge have been shown to promote compliance in preventing diseases caused by viruses [18], and for this, information is an important factor. The results from the current study show that, in this initial phase of the COVID-19 pandemic, a vast majority of the respondents perceive the information from Swedish authorities as sufficient, clear and reliable. Further, our results show that 96.8% are taking the recommendations to limit infection seriously, indicating a possible intent to take action [19].

Compliance can also be driven by factors connected to the specific crisis or threat. In the case of the current pandemic, there are several intrinsic risk factors that might explain the high reported compliance. It is likely that a sense of solidarity plays a role, a message that has been transmitted to the Swedish public via media: if you get sick, you risk passing the virus to more vulnerable individuals, as well as putting stress on the healthcare system. Therefore, as discussed in Cheng et al. [20], when trying to avoid infection, you contribute to the collective efforts of limiting the pandemic. Further, with almost 75% reporting that their loved ones worried about them, and perhaps appealed for compliance, peer influence also comes in to play. Our results also show that the confidence in recommendations from Swedish authorities is high, yet another factor that is important for compliance [21]. More than 60% reported worries about their own health, which promotes compliance [22,23]. However, although almost everyone has changed their daily routines, and, e.g., avoid public gatherings, almost one fifth report shopping for groceries as usual, indicating a discrepancy between the will and intention to comply, and the actual behaviour. For a deeper understanding of this, further research using behavioural science theory is required [24,25].

Although we do not know anything about the participants’ previous mental health status, our results showed that somewhere between more than one fifth and half of the participants reported that the COVID-19 pandemic has contributed to mental health problems in terms of, e.g., sleeping problems and symptoms of depression, in accordance to a recent review [26]. In our sample, single people were not more prone to worry about their health, although it has been shown that compared to married individuals, the widowed, divorced/separated and never married had a higher death rate [27]. However, in line with previous research [15,16], single people and women reported significantly greater numbers in mental health symptoms. This could be compared to a Swedish national survey [28], where one third of the women and one fifth of the men suffer from feeling depressed and almost half of the women and one third of the men report sleeping problems. However, our questionnaire question implied a relation to the COVID-19 pandemic, hence our results could be interpreted as being on top of the mental health burden. When we distributed our questionnaire, social distancing had been recommended for about four weeks, and more than a half (63.2%) of the participants reported that they spend all of their time at home. This does not necessarily mean not going outside at all. However, half of the respondents reported that they felt depressed due to the isolation, which could be a symptom of major depression [29]. Social distancing has also forced people to give up on physical activities and sports, activities that have been shown to prevent mental health problems [30]. Resuming activities after breaking the habit can be hard, and for an elderly person who rapidly loses ability when inactive, this can be especially demanding. Physical distancing or isolation does not automatically mean no contact with family and friends online or via telephone calls. Perceived social support via contemporary communication technologies can counteract and decrease the feeling of isolation [31].

### 4.2. Strengths and Limitations

To the best of our knowledge, this study is one of first studies to investigate the impact of the COVID-19 pandemic on the elderly in Sweden. The current questionnaire was distributed about four weeks after the recommendations for the elderly were announced, and the collected data describe an early stage of the pandemic. However, there are limitations associated with our study. Firstly, due to the cross-sectional nature of this study, we cannot make conclusions on a causal relationship of the COVID-19 pandemic on mental health. However, in the questionnaire, the respondents were asked to reply in relation to the current pandemic, and that is how the results have been interpreted. Secondly, since we used two digital channels (Facebook and the Swedish National Pensioners Organisation) due to shortage of time, there is the risk of self-selection bias. The generalisation of our findings is limited to a population among elderly people that uses the Internet, approximately 87 % in Sweden [32], which could also possibly be linked to a higher socioeconomic status. Missing those not using the Internet and groups with lower socioeconomic status could potentially affect the results, e.g., in terms of overestimating the level of compliance. Another limitation is due to the absence of a full lock-down in Sweden. Instead, a voluntary quarantine including avoiding all social interactions outside the household for people 70 and older was recommended; this needs to be taken into consideration when comparing the findings to other countries. Nevertheless, our study gives an important snapshot of the lives of elderly people in Sweden during a very difficult time.

## 5. Conclusions

At this point, we do not know the full extent of the ongoing pandemic, either in terms of duration or in terms of losses. The results from this study indicate that most elderly people perceive information as sufficient and clear, and that compliance to recommendations and trust in government authorities are high. However, elderly people are not a homogenous group, and there are individuals reporting that they go about their daily activities (like grocery shopping) as usual. Further, for vulnerable individuals, i.e., those who do not have the ability to assimilate and act on information, the results might look different, and a lack of support systems can increase the risk of noncompliance as well as mental health problems. In this study, half of the participants state that forced isolation and social distancing affect their mental health. This raises the concern of long-term effects, and highlights an area in need of further investigation.

## Figures and Tables

**Table 1 ijerph-17-05380-t001:** Demographic characteristics of the study population, number and percentages (n, (%)).

Variables		n (%)
Sex n = 1833 (98.9%)	Women	1272 (69.4)
	Men	561 (30.6)
Age groups n = 1854 (100.0%)	69–74	1058 (57.1)
	75–99	796 (42.9)
Civil status n = 1842 (99.4%)	Not single	1096 (56.9)
	Single	790 (43.1)

The percentage does not sum to 100 due to missing cases.

**Table 2 ijerph-17-05380-t002:** Perception of information based on gender (number, percentage and Chi^2^-analysis).

Variables	Total n (%)	Women n (%)	Men n (%)	Chi^2^-Value	*p*-Value
*The information concerning the COVID-19 pandemic…*					
*…is sufficient* yes no	180 81662 (91.9) 146 (8.1)	1161 (92.7) 91 (7.3)	501 (90.1) 55 (9.9)	3.570	0.059
*…is clear and concise* yes no	1796 1563 (87) 233 (13)	1102 (88.2) 148 (11.8)	461 (84.4) 85 (15.6)	4.677	**0.031**
*...from government authorities is reliable* *yes* *no*	1797 1556 (86.6) 241 (13.4)	1086 (87.2) 159 (12.8)	470 (85.1) 82 (14.9)	1.430	0.232
*…from media is reliable* yes no	1790 912 (50.9) 878 (49.1)	662 (53.3) 580 (46.7)	250 (45.6) 298 (54.4)	8.976	**0.003**

Bold text indicates significant differences (*p* < 0.05).

**Table 3 ijerph-17-05380-t003:** Perceptions concerning recommendations based on gender (number, percentage and Chi^2^-analysis).

Variables	Total	Women n (%)	Men n (%)	Chi^2^-Value	*p*-Value
*In relation to the COVID-19 pandemic…*					
*Recommendations from government authorities are clear and concise* yes no	1793 1586 (88.5) 207 (11.5)	1120 (90.2) 122 (9.8)	466 (84.6) 85 (15.4)	11.736	**0.001**
*I take the recommendations from government authorities very seriously* yes no	1814 1756 (96.8) 58 (3.2)	1232 (97.5) 31 (2.5)	524 (95.1) 27 (4.9)	7.414	**0.006**
*I know how to act to avoid infection* yes no	1807 1776 (98.3) 31 (1.7)	1244 (99.0) 12 (1.0)	532 (96.6) 19 (3.4)	14.115	**0.001**
*I have changed my daily routines* yes no	1806 1723 (95.4) 83 (4.6)	1202 (95.9) 51 (4.1)	521 (94.2) 32 (5.8)	2.578	0.108
*I avoid public gatherings* yes no	1803 1787 (99.1) 16 (0.9)	1243 (99.2 10 (0.8)	544 (98.9) 6 (1.1)	0.373	0.542
*I spend all of my time at home* yes no	1802 1138 (63.2) 664 (36.8)	826 (66.0) 426 (34.0)	312 (56.7) 238 (43.3)	14.042	**0.000**
*I do grocery shopping as usual* yes no	1786 319 (17.9) 1467 (82.1)	180 (14.5) 1060 (85.5)	139 (25.5) 407 (74.5)	30.935	**0.000**
*I do grocery shopping, but at hours when I think no one is there* yes no	1795 861 (48.0) 934 (52.0)	576 (46.1) 673 (53.9)	285 (52.2) 261 (47.8)	5.629	**0.018**

Bold text indicates significant differences (*p* < 0.05).

**Table 4 ijerph-17-05380-t004:** Reported mental health based on gender (number, percentage and Chi^2^-analysis).

Variables	Total n (%)	Women n (%)	Men n (%)	Chi^2^-Value	*p*-Value
*In relation to the COVID-19 pandemic…*					
*I worry about my health* yes no	1797 1092 (60.8) 705 (39.2)	779 (62.3) 471 (37.7)	313 (57.2) 234 (42.8)	4.149	**0.042**
*I worry about my loved ones* yes no	1789 1427 (79.8) 362 (20.2)	997 (80.1) 248 (19.9)	430 (79.0) 114 (21.0)	0.252	0.616
*My loved ones worry about me* yes no	1780 1324 (74.4) 456 (25.6)	921 (74.3) 318 (25.7)	403 (74.5) 138 (25.5)	0.005	0.944
*I worry about the economic development* yes no	1785 1168 (65.4) 617 (34.6)	853 (68.6) 390 (31.4)	315 (58.1) 227 (41.9)	18.419	**0.000**
*The isolation makes me feel bad* yes no	179 8896 (49.8) 902 (50.2)	693 (55.4) 559 (44.6)	203 (37.2) 343 (62.8)	50.220	**0.000**
*I have trouble sleeping* yes no	1791 420 (23.5) 1371 (76.5)	334 (26.8) 912 (73.2)	86 (15.8) 459 (84.2)	25.678	**0.000**
*I feel depressed* yes no	179 8697 (38.8) 1101 (61.2)	557 (44.4) 697 (55.6)	140 (25.7) 404 (74.3)	55.788	**0.000**
*I have difficulties concentrating* yes no	1775 411 (23.2) 1364 (76.8)	332 (26.9) 903 (73.1)	79 (14.6) 461 (85.4)	31.702	**0.000**
*I worry about the consequences to society* yes no	1794 1505 (83.9) 289 (16.1)	1077 (86.1) 174 (13.9)	428 (78.8) 115 (21.2)	14.808	**0.000**

Bold text indicates significant differences (*p* < 0.05).

**Table 5 ijerph-17-05380-t005:** Self-reported mental health based on civil status (number, percentage and Chi^2^-analysis).

Variables	Total n (%)	Not Single n (%)	Single n (%)	Chi^2^-Value	*p*-Value
*In relation to COVID-19 pandemic…*					
*I worry about my health* yes no	1806 1099 (60.9) 707 (39.1)	654 (61.4) 412 (38.6)	445 (60.1) 295 (39.9)	0.271	0.603
*I worry about my loved ones* yes no	179 81,434 (79.8) 364 (20.2)	885 (83.0) 181 (17.0)	549 (75.0) 183 (25.0)	17.291	**0.000**
*My loved ones worry about me* yes no	17 881,330 (74.4) 458 (25.6)	789 (74.5) 270 (25.5)	541 (74.2) 188 (25.8)	0.019	0.889
*I worry about the economic development* yes no	1793 1173 (65.4) 620 (34.6)	679 (64.2) 378 (35.8)	494 (67.1) 242 (32.9)	1.592	0.207
*The isolation makes me feel bad* yes no	1807 904 (50.0) 903 (50.0)	482 (45.3) 583 (54.7)	422 (56.9) 320 (43.1)	23.599	**0.000**
*I have trouble sleeping* yes no	1799 422 (23.5) 1377 (76.5)	220 (20.7) 841 (79.3)	202 (27.4) 563 (72.6)	10.675	**0.001**
*I feel depressed* yes no	1806 701 (38.8) 1105 (61.2)	356 (33.5) 707 (66.5)	345 (46.4) 398 (53.6)	30.849	**0.000**
*I have difficulties concentrating* yes no	1783 413 (23.2) 1370 (76.8)	203 (19.3) 847 (80.7)	210 (28.6) 523 (71.4)	21.049	**0.000**
*I worry about the consequences to society* yes no	1803 1512 (83.9) 291 (16.1)	896 (84.3) 167 (15.7)	616 (83.2) 124 (16.8)	0.353	0.552

Bold text indicates significant differences (*p* < 0.05).

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
