# Peer review of "Compliance to Recommendations and Mental Health Consequences among Elderly in Sweden during the Initial Phase of the COVID-19 Pandemic—A Cross Sectional Online Survey"

_ijerph, 2020, doi:10.3390/ijerph17155380_

Round 1

Reviewer 1 Report

Comments to the authors:

In my opinion, this manuscript entitled “Compliance to recommendations and mental health consequences among elderly in Sweden during the initial phase of the COVID-19 pandemic - a cross sectional online survey” is a very interesting article that cover an extremely current topic treating it from an interesting perspective that concerns the effects on the elderly population deriving from the compliance to the pandemic control measures applied in Sweden. The rationale is well presented in the introduction, the methodology is adequate and the results well described. Then, overall the manuscript is well written and the Discussion section sheds an adequate perspective in the context of the author's work. Consequently, I have only few  suggestions/observations regarding some points which need to be addressed:

Introduction

  • Instead of coronavirus I think it would be better write SARS-CoV2;
  • The authors correctly stated that “Early on, higher age was identified as a primary risk factor for death caused by the coronavirus”. I suggest to describe also the other fundamental risk factors such as comorbidity;

Materials and methods

  • In my opinion, even if there is a reference to the Appendix 1, a more detailed description (how many items are included? What are the investigated areas? Has the questionnaire been validated? or in any case has it been tested initially in a small group of elderly people to verify its adequacy, understanding by the reader?) would be helpful for a better comprehension by the reader;

Discussion

  • Lines 187-188: Cheng, Lam [19] please correct “Chen and Lam”;

Author Response

Dear Editor and Reviewers,

We hereby resubmit the article “Compliance to recommendations and mental health consequences among elderly in Sweden during the initial phase of the COVID-19 pandemic - a cross sectional online survey”.

We like to thank the reviewers for reading the manuscript and contributing with valuable comments, which we believe have improved the article.

Response på comments are presented below. Revisions are highlighted in grey.

Best Regards/The Authors

Reply to reviewer 1

Introduction

Comment: Instead of coronavirus I think it would be better write SARS-CoV2;

Response: Thank you for this valid remark, changes have been made accordingly in Introduction, row 30-36: “In December 2019, a novel coronavirus labeled SARS-CoV2, causing severe respiratory illness, appeared in Wuhan, China. During the first months of 2020, the virus spread and on 11 March 2020, the World Health Organization (WHO) declared the outbreak a pandemic [1]. Early on, higher age was identified as a primary risk factor for death caused by the SARS-CoV2 [3]. Compared to most countries, Sweden has chosen a different path in efforts to minimise the consequences of the SARS-CoV2 outbreak, i.e. the COVID-19 pandemic.”

Comment: The authors correctly stated that “Early on, higher age was identified as a primary risk factor for death caused by the coronavirus”. I suggest to describe also the other fundamental risk factors such as comorbidity;

Response: Thank you for noticing the lack of describing potential risk factors for SARS-CoV2 infections other than higher age. A section has been added in Introduction, row 33: “Early on, diabetes and hypertension comorbidities were identified as risk factors for severe infection [2], and higher age a primary risk factor for death caused by the SARS-CoV2 [3].”

Materials and methods

In my opinion, even if there is a reference to the Appendix 1, a more detailed description (how many items are included? What are the investigated areas? Has the questionnaire been validated? or in any case has it been tested initially in a small group of elderly people to verify its adequacy, understanding by the reader?) would be helpful for a better comprehension by the reader;

Response: The items/questionnaire have not been validated, but initially we tested the questionnaire on a smaller a group of elderly. After that, we reconstructed some of the questions so that they were conceptually sound. We have added a para in the method section [page 2, row 83-85 “Initially we tested the questions on a group of elderly in order to test its conceptual validity. After that, we changed some of the questions and wording”. We have further added additional information about the questionnaire [page 2, row 73-76: “The questionnaire included 5 questions with 5 to 10 items in each question about perceptions of information concerning COVID-19, how they perceive and comply with the recommendations made towards the risk group to which they belong, as well as how they estimate their mental health status during the pandemic situation”. Thank you for noticing this.

Discussion

Comment: Lines 187-188: Cheng, Lam [19] please correct “Chen and Lam”;

Response: Thank you for noticing the incorrect reference. As the article has three authors we assume that the correct reference is Cheng et al. In Result discussion, row 194 the text has been revised: Therefore, as discussed in Cheng et al [19], when trying to avoid infection, you contribute to the collective efforts of limiting the pandemic.”

Reviewer 2 Report

I assessed the manuscript by Gustavsson and Beckman regarding the potential impact of recommendation to avoid SARS-CoV-2 infection on metal health of elderly in Sweden. The authors designed a cross-sectional web-based survey to assess the topic which is a reliable tool for the aim of the study. Overall, the topic is of interest considering the current epidemic situation and the potential detrimental effects of lockdown on mental health. Elderly people could be considered a special population of more frail people especially at risk for COVID-19 and consequently it seems to be reasonable that the main objective of the research focus on these patients. Overall, the manuscript is well written and the methodology is clearly described. Nevertheless, there are some limitations that should be acknowledge. In particular regarding the specific setting (the absence of lock-down) which is specific for Sweden and thus not generalizable to other countries. Moreover, the survey was disseminated trough two specific channel and thus it is possible that the respondents could not represent the overall elderly Swedish population. In the end, important demographic information that could impact on the results interpretation were not reported.

Major comments

Demographic information:

  • Did the authors have information regarding the educational level of the respondents? This is an important variable that could act as confounders in the interpretation of the results. Due to the nature of the survey (on-line) it is possible that people with a higher educational level could have accessed the questionnaire.
  • Did the authors have information regarding other family members that could act as caregivers or as a support in this situation?
  • How many patients were still working? Were all retired?

Which was the national coverage of the survey? It could be helpful to have a figure with the distribution of the questioner in the specific Swedish regions. This could help to interpret were the respondents lives (i.e. Stockholm or small towns).

Minor comments

  • Page 3 methods line 103-114 the misprint must be cut.

Author Response

Reply to reviewer 2

General comments

Comment: In particular regarding the specific setting (the absence of lock-down) which is specific for Sweden and thus not generalizable to other countries.

Response: We agree that this is an aspect that effects the generalization worth noticing. A sentence have been added in the 238-242 “Another limitation is due to the absence of a full lock-down in Sweden. Instead a voluntary quarantine including avoiding all social interactions outside the household for people 70 and older were recommended, this needs to be taken in to consideration when comparing the findings to other countries.”

Major comments

Demographic information:

Comment: Did the authors have information regarding the educational level of the respondents? This is an important variable that could act as confounders in the interpretation of the results. Due to the nature of the survey (on-line) it is possible that people with a higher educational level could have accessed the questionnaire.

Response: As the reviewer points out, higher education (or higher socioeconomic status) could contribute to self-selection bias. However, for this population, higher education is not necessarily equal to higher socioeconomic status since it was possible for members of this generation to work their way up the hierarchy without formal merits. Nevertheless, self-selection is correctly a problem in our study and we address this in “strengths and limitations”, where we make clear that the findings in our study is not to be generalized to all of the Swedish elderly population, neither is it possible to fully compare them with other countries. In order to emphasize this even more, we have added a section in the discussion. In Strengths and limitations, row 234-243: “The generalisation of our findings is limited to a population among elderly people that uses the Internet, approximately 87 % in Sweden [32], which could also possibly be linked to a higher socioeconomic status. Missing those not using the Internet and groups with lower socioeconomic status having the prerequisites to participate could potentially affect the results, e.g. in terms of overestimating the level of compliance. Another limitation is due to the absence of a full lock-down in Sweden. Instead a voluntary quarantine including avoiding all social interactions outside the household for people 70 and older were recommended, this needs to be taken in to consideration when comparing the findings to other countries. Nevertheless, our study gives an important snapshot of the lives of elderly people in Sweden during a very difficult time.”

Comment: Did the authors have information regarding other family members that could act as caregivers or as a support in this situation?

Response: We agree that this is important information, but unfortunately it is not included in this study.

Comment: How many patients were still working? Were all retired?

Response: We do not have information about whether they are working or not. However, in Sweden the general retirement age is 67 years of age. About 18 % of elderly between 65 and 74 are still, though not full-time, in the labor force.

Comment: Which was the national coverage of the survey? It could be helpful to have a figure with the distribution of the questioner in the specific Swedish regions. This could help to interpret were the respondents lives (i.e. Stockholm or small towns).

Response: This is of course interesting and important information, however we did not include this information since we wanted to keep the questionnaire as anonymous as possible. However, in hindsight this information would have been beneficial to have had.

Minor comments

Comment: Page 3 methods line 103-114 the misprint must be cut.

Response: Correct, the last section of paragraph 2.4. Analyses are instructions that we missed to delete.

Round 2

Reviewer 2 Report

I want to thank the authors for their revision and reply.